# Gender specific differences in COVID-19 knowledge, behavior and health effects among adolescents and young adults in Uttar Pradesh and Bihar, India

Jessie Pinchoff[1]*, KG Santhya[2], Corinne White[1], Shilpi Rampal[2], Rajib Acharya[2], Thoai D. Ngo[1]

1 Population Council, One Dag Hammarskjold Plaza, New York, NY, United States of America, 2 Population Council, India Habitat Centre, New Delhi, Delhi, India

* jpinchoff@popcouncil.org

**Data Availability Statement:** The data are accessible via Dataverse (https://doi.org/10.7910/DVN/8ZVOKW).

## Abstract

On March 24, 2020 India implemented a national lockdown to prevent spread of the novel Coronavirus disease (COVID-19) among its 1.3 billion people. As the pandemic may disproportionately impact women and girls, this study examines gender differences in knowledge of COVID-19 symptoms and preventive behaviors, as well as the adverse effects of the lockdown among adolescents and young adults. A mobile phone-based survey was implemented from April 3–22, 2020 in Uttar Pradesh and Bihar among respondents randomly selected from an existing cohort study. Respondents answered questions related to demographics, COVID-19 knowledge, attitudes, and preventive behaviors practiced, and impacts on social, economic and health outcomes. Descriptive analyses and linear probability regression models were performed for all participants and separately for men and women. A total of 1,666 adolescents and young adults (18–24 years old) were surveyed; 70% were women. While most participants had high awareness of disease symptoms and preventive behaviors, there was variation by gender. Compared to men, women were seven percentage points (pp) less likely to know the main symptoms of COVID-19 (coeff = -0.071; 95% confidence interval: -0.122 - -0.021). Among women, there was variation in knowledge by education level, urban residence, and household wealth. Women were 22 pp less likely to practice key preventive behaviors compared to men (coeff = -0.222; 95% CIL -0.263, -0.181). Women were also more likely to report recent depressive symptoms than men (coeff = 0.057; 95% CI: 0.004, 0.109). Our findings underscore that COVID-19 is already disproportionately impacting adolescent girls and young women and that they may require additional targeted, gender-sensitive messaging to foster behavior change. Gender-sensitive information campaigns and provision of health services must be accessible and provide women and girls with needed resources and support during the pandemic to ensure gains in public health and gender equity are not lost.

**Funding:** The initial UDAYA cohort was funded by the Bill and Melinda Gates Foundation and Packard Foundation. No additional funds were received for the COVID-19 survey.

**Competing interests:** The authors have declared that no competing interests exist.

## Introduction

To control the spread of the novel Coronavirus disease (COVID-19), the Indian government swiftly instituted a shutdown of international borders and a stay-at-home order on March 24, 2020 [1]. Such 'lockdown' policies to prevent the spread of COVID-19 originated in high-income countries and China; little primary research has explored the potential unintended consequences in countries including India characterized by densely populated urban slums, a highly mobile population, high proportions of informal sector workers, and stark variation in poverty levels [2]. Despite rising cases, the lockdown was lifted on June 8, 2020 to begin a phased reopening. As of August 2020, India surpassed 2.3 million cases of COVID-19, the third highest case load after the United States and Brazil [3].

Historically, epidemics and humanitarian crises have disproportionately impacted the most vulnerable, including women and girls [4]. Entrenched inequalities in access to education, job opportunities, and healthcare often leave women inadequately equipped to effectively protect themselves and their families against infection during an outbreak, and they are also more likely to bear secondary negative effects of prolonged crises, such as economic insecurity or challenges accessing essential health services [5]. Existing gender disparities in India may be exacerbated or reinforced by the pandemic and are likely to affect women's ability to make informed decisions about adopting behaviors that mitigate risk of COVID-19.

Prevention campaigns and behavior change communication interventions across various media, including a government-run mobile app ("Aarogya Setu") that sends automated messages, are informing the public about COVID-19 symptoms, risk factors, and promoting preventive behaviors such as handwashing, social distancing, and wearing masks in India. To date, there is little to no research tracing how COVID-19 messages are reaching men and women or which sub-groups are adopting these behavioral recommendations. However, a rapid situational assessment in the South Asia region (not including India) suggests that women are less likely than men to have received COVID-19 information [6]. Moreover, literacy, internet usage and smartphone ownership is lower among women compared to men in India [7–9]. Accessing and understanding health promotion messages increases knowledge, which needs to be accompanied with structural facilitators and access to resources to adopt promoted preventive behaviors (e.g., making soap and water available for handwashing) [10–12]. These gender gaps may result in lower adoption of promoted health behaviors and increased risk of infection for women and girls.

The worsening COVID-19 pandemic in India is causing prolonged social and economic disruptions that are yielding unintended consequences including economic and food insecurity, and challenges in accessing healthcare. Challenges in accessing essential health services may lead to increases in other adverse health outcomes, from vaccine preventable diseases to poor birth outcomes and malnutrition [13,14]. This often disproportionately harms women who may require healthcare themselves and are also often responsible for taking care of their family's health needs. Potential reasons for these challenges may include inability to pay clinic fees as COVID-19 related economic insecurity persists, mobility challenges, or fear of seeking care due to stigma or concerns about COVID-19 infection at the facility. Indeed, compared to March 2019, March 2020 data from the Indian National Health Mission showed marked reductions in indicators of regular health system usage [2].

In addition to physical health, lockdowns may exacerbate household stress, contributing to increases in sexual and gender-based violence (SGBV) and poor mental health symptoms [15,16]. While psychological distress increases generally during crises, experience of depressive symptoms is more common among women compared to men [17]. In addition to gender, a recent study also found that adolescents and younger adults (<25 years), those that had lost

employment, and/or lacked formal education were more likely to experience depressive symptoms as a result of the pandemic's effects [18]. Relatedly, stress and ongoing lockdowns have been linked with violence against women, as in past humanitarian crises [19]. Some countries reported increases in SGBV during COVID-19 lockdowns [15,20]. Concerns around these secondary health and well-being effects are significant.

As India is home to the largest population of adolescents and young adults of any country worldwide, understanding the impact of the pandemic on this important age cohort will also be critical. In the age- and gender-stratified settings of India, prevailing gender disparities and traditional gender norms affect health and well-being of adolescents and young people disproportionately. However, little is known regarding the experience during the COVID-19 pandemic of Indian adolescent girls and young women compared to men. A cross-sectional mobile phone-based survey of households in Uttar Pradesh (UP) and Bihar was carried out four to six weeks after lockdown was imposed. This analysis highlights the gender specific variation in COVID-19 knowledge and practice of preventive behaviors, and mental health effects among a cohort of adolescent and young adults. Findings from this study can inform the development of social service programs and education campaigns to ensure that adolescent and young women have access to tailored information and resources during this protracted crisis to ensure development and equity gains are not lost.

## Methods

### Sampling strategy

A rapid telephone survey was conducted with a sample of participants drawn from an existing Population Council cohort study of adolescents and young adults. Understanding the Lives of Adolescents and Young Adults (UDAYA) is a state-level representative longitudinal study of adolescent girls and boys (aged 10–19) in rural and urban settings in Bihar (n = 10,433) and UP (n = 10,161), with baseline conducted in 2015–2016 and endline in 2018–19. The original UDAYA study objectives were to better understand adolescents' acquisition of assets and their transition from adolescence to adulthood [21,22]. UDAYA researchers used the 2011 Indian Census to create a systematic, multi-stage sampling frame for the selection of 150 primary sampling units (PSU) in each state, with an equal breakdown between urban and rural areas. UDAYA was designed to provide estimates for five categories of adolescents, namely unmarried younger boys and girls aged 10–14, unmarried older boys and girls aged 15–19, and married older girls aged 15–19 that represent each state [21,22].

UDAYA households eligible for inclusion in the COVID-19 survey were those in which we interviewed a 15-19-year-old boy or girl in 2015–16. Phone numbers were available for 9,771 of such UDAYA participants– 2,437 boys and men and 7,334 girls and women. We randomly sampled households for the mobile phone survey from this list of telephone numbers, stratified by gender. The enumerators contacted telephone numbers belonging to 5,520 UDAYA participants– 1,512 boys and men and 4,008 girls and women–attempting each number up to 3 times and completing about 10 interviews per day. Of those attempted, 51% of telephone numbers were no longer functional (of UDAYA participants, 44% of boys and men and 53% of girls and women). Of numbers we successfully reached, 5% of respondents refused to participate in the study. Overall, participants in the COVID-19 study had slightly higher educational attainment, were slightly more urban, and had slightly higher household wealth compared to the source cohort. The characteristics of the UDAYA baseline cohort compared to those who were enrolled in the COVID-19 mobile-phone survey is summarized in a S1 Table.

## Mobile phone questionnaire

Participants were contacted via mobile phone to remove the risk to field staff and participants of COVID-19 infection. After verbal consent for participation, a short questionnaire lasting no longer than 30 minutes was administered. The questionnaire included questions regarding basic demographics, awareness of COVID-19 or coronavirus, knowledge of symptoms, risk groups and transmission, perceived risk, COVID-19 prevention behaviors, and fears or concerns regarding the outbreak. Questions assessing household and individual needs under the government lockdown were also included. In the survey participants self-reported their sex as male or female; throughout this paper we will refer to respondents as men and women to illustrate that our analysis reports how the pandemic impacts gender (the socially constructed characteristics of men and women) not biological sex.

## Ethical review

We received expedited ethical approval from the Population Council's Institutional Review Board (IRB) by meeting criteria for research conducted during COVID-19. The IRB permitted data collection with participants with previous consent from existing cohort studies, provided the research is aligned with national mitigation efforts. The UDAYA study protocol originally received IRB approval in 2015 for longitudinal data collection. Participants were told they could terminate the study at any time or skip any sections. No incentives were offered for taking part in the study.

## Data management and analysis

The survey responses were entered in mini laptops using instruments developed with CSPro 7.1 and exported to Stata v15 for analysis. Each household had a unique ID number, and all personally identifiable information was removed to ensure confidentiality.

Two summary outcome variables were created. First, participants who correctly identified all three COVID-19 symptoms (fever, cough and difficulty in breathing) were considered to have correct knowledge (dichotomous variable). Participants who reported implementing all four preventive behaviors (staying home more, wearing a mask, washing hands/using sanitizer, and staying 2m apart) were categorized as implementing the four main preventive behaviors (dichotomous variable). Depressive symptoms, as measured by reporting feeling lonely, depressed or irritable during the lockdown, was collected as a dichotomous variable. To control for household wealth, we created a proxy variable constructed from the presence of four basic amenities: safe drinking water, electricity, toilet facility and safe cooking fuel. Educational attainment was categorized into three levels, with grade 8 indicating completion of primary education and grade 10 indicating completion of secondary education. Religion was categorized as Hindu or Muslim (dichotomous variable), with 9 indicating 'other' and excluded from models. Lastly, caste was categorized as scheduled caste/tribe (SC/ST), other backward castes (OBC) and general (neither SC/ST nor OBC); these designations, as provisioned in the Indian constitution, are used to identify marginalized groups in the population. Only women were asked if they had experienced any violence in the home in the last 15 days under lockdown.

All survey responses were tabulated by gender and tested for statistical significance ($p < 0.05$) using chi-square tests. We implemented linear probability regression models based on three outcomes of interest. First, knowledge of all three key symptoms of COVID-19. Second, practicing all four of the key preventive behaviors. The third outcome was self-reported experience of loneliness, depression, or irritability (dichotomous variable) in the previous seven days used to define experience of depressive symptoms. Three separate linear probability

regression models were constructed for each of the three outcome variables, first for the full set of respondents and then stratified by gender.

## Results

A total of 1,666 adolescents and young adults (18–24 years) previously enrolled in the UDAYA study were surveyed. Of these, 70% were women, over half had completed 10+ years of education (72%) and nearly half resided in urban areas (47%) (Table 1). Fewer women (40%) than

**Table 1. Demographics and COVID-19 related outcomes of interest tabulated by gender.**

| | Men | Women | Total | p-value |
|---|---|---|---|---|
| | N = 506 | N = 1,160 | N = 1,666 | |
| **Demographic/Household characteristics** | | | | |
| **Age group** | | | | 0.058 |
| 18–19 years | 88 (17%) | 160 (14%) | 248 (15%) | |
| 20–24 years | 418 (83%) | 1,000 (86%) | 1,418 (85%) | |
| **Religion** | | | | <0.001 |
| Hindu | 432 (85%) | 911 (79%) | 1,343 (81%) | |
| Muslim | 68 (13%) | 246 (21%) | 314 (18%) | |
| Other | 6 (1%) | 3 (0%) | 9 (1%) | |
| **Completed years of education** | | | | <0.001 |
| 0–7 years | 25 (5%) | 195 (17%) | 220 (13%) | |
| 8–9 years | 59 (12%) | 191 (16%) | 250 (15%) | |
| 10 and above years | 422 (83%) | 774 (67%) | 1,196 (72%) | |
| **Caste** | | | | 0.785 |
| General caste | 108 (21%) | 262 (23%) | 370 (22%) | |
| Other backward caste (OBC) | 268 (53%) | 615 (53%) | 883 (53%) | |
| Scheduled caste/tribe | 130 (26%) | 283 (24%) | 413 (25%) | |
| **Current place of residence** | | | | <0.001 |
| Urban (vs Rural) | 274 (54%) | 502 (43%) | 776 (47%) | |
| **Have four key amenities[1]** | | | | 0.014 |
| Yes (vs No) | 180 (36%) | 342 (29%) | 522 (31%) | |
| **COVID-19 Outcomes of Interest** | | | | |
| **Mental Health: have you felt depressed, lonely or irritable under lockdown?** | | | | |
| Never | 972 (58%) | 321 (63%) | 651 (56%) | |
| Sometimes | 578 (35%) | 159 (31%) | 419 (36%) | |
| Most of the time | 116 (7%) | 26 (5%) | 90 (8%) | 0.011 |
| *Knowledge and behaviors* | | | | |
| Knows all 3 top symptoms[2] | 266 (53%) | 463 (40%) | 729 (44%) | <0.001 |
| Reports practicing all 4 main preventive measures[3] | 199 (39%) | 158 (14%) | 357 (21%) | <0.001 |
| *Economic and health access effects* | | | | |
| Self or household member lost job/income source due to COVID-19 | 274 (54%) | 788 (68%) | 1,062 (64%) | <0.001 |
| Among women who required each health service but could not access it: | | | | |
| Antenatal care | - | 138 (12%) | - | - |
| Family planning | - | 239 (21%) | - | - |
| Child immunization | - | 433 (37%) | - | - |
| Nutrition | - | 595 (51%) | - | - |

Notes

[1] Includes source of light i.e. electricity, source of water i.e., improved water, source of clean fuel i.e. LPG/bio-gas and type of toilet facility i.e., own/public flush toilet

[2] Three main symptoms are fever, cough, and difficulty breathing

[3] Four main behaviors are stay home unless urgent, keep 2m apart from others, wear a mask, and wash hands/use sanitizer

**Table 2. Linear probability model of factors associated with <u>knowledge</u> of all 3 main COVID-19 symptoms (fever, cough, difficulty breathing), stratified by gender.**

| | (1) | (2) | (3) |
|---|---|---|---|
| VARIABLES | Model 1: All | Model 2: Men | Model 3: Women |
| | | | |
| Women (vs men) | -0.069** | NA | NA |
| | (-0.122 - -0.021) | | |
| Muslim (vs Hindu) | -0.047 | -0.067 | -0.049 |
| | (-0.109–0.015) | (-0.198–0.064) | (-0.120–0.023) |
| Educational attainment (0–7 years = REF) | | | |
| 8–9 years | 0.059 | -0.102 | 0.088 |
| | (-0.028–0.146) | (-0.336–0.132) | (-0.006–0.182) |
| 10+ years | 0.242** | 0.136 | 0.250** |
| | (0.171–0.314) | (-0.070–0.342) | (0.173–0.328) |
| Caste (OBC = REF) | | | |
| General Category | 0.069* | 0.138* | 0.040 |
| | (0.010–0.128) | (0.026–0.251) | (-0.029–0.110) |
| Scheduled Caste/ Tribe | 0.022 | 0.044 | 0.014 |
| | (-0.035–0.080) | (-0.061–0.149) | (-0.056–0.083) |
| Age 20–24 (vs 18–19 years) | 0.008 | 0.020 | 0.001 |
| | (-0.056–0.073) | (-0.093–0.133) | (-0.078–0.080) |
| Rural (vs urban) | -0.067* | -0.045 | -0.077* |
| | (-0.123 - -0.011) | (-0.146–0.055) | (-0.146 - -0.009) |
| Household has all 4 amenities | 0.111** | 0.104 | 0.109** |
| | (0.049–0.172) | (-0.004–0.211) | (0.033–0.185) |
| Bihar (vs UP) | -0.039 | 0.017 | -0.061 |
| | (-0.091–0.013) | (-0.082–0.116) | (-0.123–0.001) |
| Observations | 1,666 | 506 | 1,160 |
| R-squared | 0.095 | 0.073 | 0.092 |

CI in parentheses

** p<0.01

* p<0.05

men (53%) knew the main symptoms of COVID-19 and fewer women than men practiced key preventive behaviors such as staying home unless it is urgent and wearing a mask (Table 1). Fewer women reported doing all prevention behaviors (14% vs 39% of men). A greater proportion of women respondents reported experience of depressive symptoms.

In the full model, women were less likely than men to know COVID-19 symptoms (coeff = -0.069; 95% CI: -0.122 - -0.021) (Table 2). The model was then stratified by gender (men- and women-only models). For the men-only model, there were no key characteristics associated with more or less knowledge of symptoms, except that those in the general caste category were 14 pp more likely to know the symptoms compared with those in the OBC category (coeff = 0.138; 95% CI: 0.026–0.251). In the women-only model, several characteristics were associated with having more knowledge of key symptoms. Women who had completed 10 + years of education were 25 pp more likely to know the symptoms compared with those only having zero to seven years of education (coeff = 0.250; 95% CI: 0.173–0.328); relatedly, women residing in households with key amenities were much more likely to know the symptoms

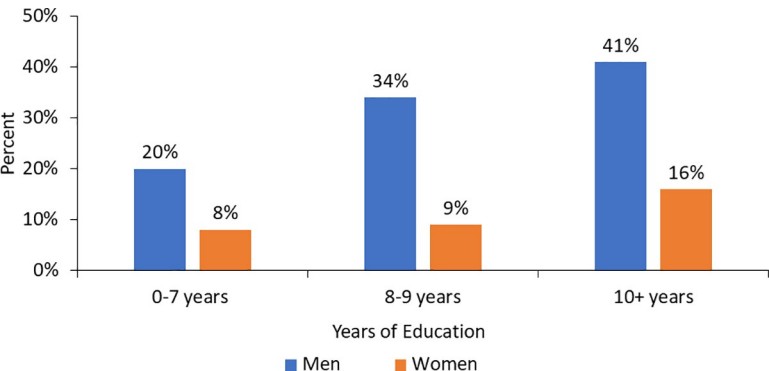

**Fig 1. Proportion of respondents that practice all four the key preventive behaviors, by gender and educational attainment.**

(coeff = 0.109; 95% CI: 0.033–0.185). Women living in rural areas had lower knowledge of the symptoms.

Fig 1 highlights the education and gender differences in reportedly practicing all four main preventive behaviors; this proportion increases across categories of educational attainment for both men and women (Fig 1). Findings also show that women respondents with secondary education (10+ years) were less likely than men respondents with less than primary education (0–7 years) to report practicing all four prevention measures.

In the full model exploring characteristics associated with doing all four prevention behaviors, women were 22 pp less likely than men to report doing all behaviors (coeff = -0.221; 95% CI: -0.263 - -0.180) (Table 3). The full model was re-run stratified by gender. Among men, several characteristics contributed to reportedly practicing all four prevention behaviors. Men who knew the top three symptoms were more likely to practice the four key preventive behaviors (coeff = 0.107; 95% CI: 0.020–0.194). Men in rural areas and in Bihar were much less likely to carry out the four behaviors. For the women-only model, the only characteristic that was associated with conducting the four behaviors was knowledge of the three main symptoms (coeff = 0.160; 95% CI: 0.119, 0.201) (Table 3).

The last model explored characteristics associated with self-reported experience of depressive symptoms. In the full model, women were 5 pp more likely to report that they were experiencing depressive symptoms compared to men (coeff = 0.052; 95% CI: -0.001, 0.104) (Table 4). When stratified by gender, among men only, household loss of employment was the only factor associated with depressive symptoms (coeff = 0.169; 95% CI 0.083, 0.254). Among women only, household loss of employment, religion, and experience of violence were significantly associated with depressive symptoms. Women belonging to the Muslim religion compared to those who identified as Hindu, were more likely to report experience of depressive symptoms (coeff = 0.084; 95% CI:0.012, 0.156). Women who reported violence in the home in the last 15 days were 30 pp more likely to report experience of depressive symptoms (coeff = 0.304; 95% CI: 0.133; 0.475).

Women reported whether they had required health services in the previous week, and if so, if they were able to access them (this question was not included for men). Most women had not required health services in the previous week. Of the types of services that were required, nutrition services and child immunization services were the most reported. Among women who sought nutrition services, 51% required but could not access them, 1% required and were able to access them. For child immunization services 37% were unable to access them, none who needed child immunization services could access them. For family planning, 76% stated

**Table 3. Linear probability model of factors associated with reporting all four main preventive behaviors are being implemented, by gender.**

| | (1) | (2) | (3) |
|---|---|---|---|
| VARIABLES | Model 1 | Model 2: Men | Model 3: Women |
| | | | |
| Women (vs men) | -0.221** | NA | NA |
| | (-0.263 - -0.180) | | |
| Knowledge of 3 key COVID symptoms | 0.134** | 0.107* | 0.160** |
| | (0.09–0.173) | (0.020–0.194) | (0.119–0.201) |
| Muslim (vs Hindu) | -0.045 | -0.090 | -0.018 |
| | (-0.096–0.005) | (-0.219–0.039) | (-0.069–0.033) |
| Educational attainment (0–7 years REF) | REF | REF | REF |
| 8–9 years | 0.009 | 0.126 | -0.008 |
| | (-0.062–0.079) | (-0.105–0.357) | (-0.075–0.059) |
| 10+ years | 0.037 | 0.187 | 0.022 |
| | (-0.022–0.096) | (-0.015–0.390) | (-0.033–0.077) |
| Caste (OBC = REF) | REF | REF | REF |
| General Category | -0.022 | -0.106 | 0.020 |
| | (-0.070–0.026) | (-0.217–0.004) | (-0.029–0.069) |
| Scheduled Caste/Tribe | -0.002 | 0.011 | -0.007 |
| | (-0.049–0.045) | (-0.092–0.115) | (-0.056–0.042) |
| Age group | -0.004 | -0.053 | 0.018 |
| | (-0.056–0.049) | (-0.165–0.059) | (-0.039–0.074) |
| Rural (vs Urban) | -0.019** | -0.147** | -0.011 |
| | (-0.074 - -0.017) | (-0.234 - -0.060) | (-0.052–0.029) |
| Bihar (vs UP) | -0.056** | -0.103* | -0.036 |
| | (-0.099 - -0.014) | (-0.201 - -0.006) | (-0.081–0.008) |
| Observations | 1,666 | 506 | 1,160 |
| R-squared | 0.128 | 0.059 | 0.066 |

CI in parentheses

** $p < 0.01$

* $p < 0.05$

they did not require this service in the previous week, of those that did, 21% could not access family planning services (84% of those with a family planning service need) (Fig 2).

## Discussion

Conducted early in the pandemic, our study identifies gender disparities in COVID-19 related knowledge and uptake of promoted preventive behaviors among young people in two states in India. Overall, women were less likely to be able to identify all three of the main COVID-19 symptoms correctly, potentially due to challenges in accessing information or receiving less accurate information of COVID-19 symptoms. Women were also less likely to be practicing the most effective prevention behaviors and they were also more likely to report symptoms of depression. Access to health services is also reportedly affected by the pandemic, with most women in need of services unable to access them, including nutrition, child immunization, family planning and antenatal care services. As of Fall 2020, the pandemic is still not under control globally, and the threat of continued infections remains; therefore, understanding the

**Table 4. Linear probability model of factors associated with self-reported experience of depressive symptoms during lockdown, by gender.**

| VARIABLES | (1) Model 1 | (2) Model 2: Men | (3) Model 3: Women |
|---|---|---|---|
| | | | |
| Women (vs men) | 0.052* | NA | NA |
| | (0.000–0.104) | | |
| Household lost employment | 0.133** | 0.169** | 0.117** |
| | (0.083–0.183) | (0.083–0.254) | (0.055–0.179) |
| Educational attainment (0–7 years REF) | REF | REF | REF |
| 8–9 years | -0.006 | 0.019 | -0.022 |
| | (-0.095–0.083) | (-0.211–0.250) | (-0.121–0.076) |
| 10+ years | 0.018 | 0.026 | 0.013 |
| | (-0.055–0.091) | (-0.174–0.226) | (-0.067–0.092) |
| Age 20–24 (vs 18–19 years) | 0.021 | -0.001 | 0.033 |
| | (-0.046–0.088) | (-0.113–0.110) | (-0.050–0.115) |
| Rural (vs urban) | -0.013 | -0.036 | 0.002 |
| | (-0.061–0.035) | (-0.121–0.049) | (-0.057–0.060) |
| Bihar (vs UP) | 0.038 | 0.059 | 0.021 |
| | (-0.015–0.092) | (-0.038–0.156) | (-0.044–0.086) |
| Muslim (vs Hindu) | 0.073** | 0.041 | 0.084** |
| | (0.011–0.135) | (-0.085–0.168) | (0.012–0.156) |
| Under lockdown, experienced any violence in the home in the last 15 days (women only) | NA | NA | 0.304** |
| | | | (0.133–0.475) |
| Observations | 1,658 | 501 | 1,157 |
| R-squared | 0.027 | 0.038 | 0.028 |

CI in parentheses

** p<0.01

* p<0.05

needs and experiences of adolescents and young adults is critical to offering resources and social support, with attention to gender.

Gender differences in accurate knowledge of key COVID-19 symptoms likely reflect young women's lower levels of educational attainment and lower media exposure, as well as lower access to mobile phones [21,22]. Among women, there was significant variation in the characteristics of who had COVID-19 information, such as higher educational attainment, urban residence, and higher economic status. These factors likely reflect higher literacy and access to information among some young women. Interestingly, no variation was observed within men, and overall, their knowledge was higher than for women. This finding is supported by available literature on past pandemics. During an outbreak of influenza A (H1N1) in India, a small study found that men had more knowledge of H1N1; this was attributed to men having more

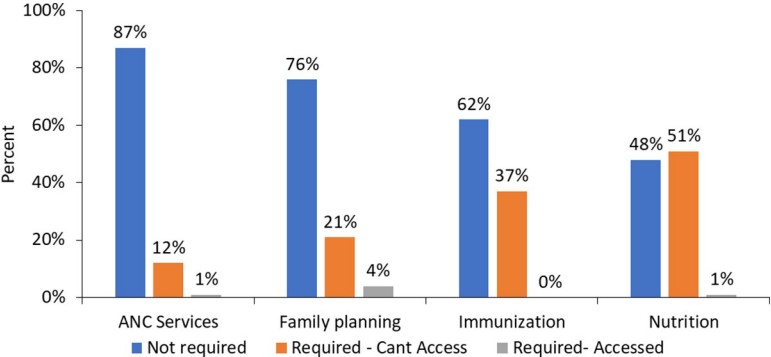

**Fig 2. Among women, number requiring health services and of those, number unable to obtain them by type of service.**

social interactions through employment and having higher literacy rates than women [23]. Higher knowledge among men may be influenced by their greater exposure to risk outside the home for work and socializing shaped by gendered social norms. A recent study from India found differential COVID-19 risk and mortality by gender, reporting that most infections are among men [24]. Our study also suggests that men have higher potential exposure but also higher knowledge of COVID-19 symptoms and prevention; gender dynamics and social norms may increase both knowledge and infection risk among men. Among women, lower adoption of promoted behaviors may also reflect the gender roles and the fact that women spend more time indoors. If women are not going outside, they may not be wearing masks or keeping 2m distance from others because they are not interacting outside the household. Knowledge was the only factor associated adoption of promoted behaviors among women; potentially there are other unmeasured characteristics that are associated with observed variation among women. To bridge this knowledge gender gap, additional research on whether and how the pandemic is reinforcing gender roles may help inform gender sensitive education campaigns via media that women can access and understand even with limited literacy.

Mental health and healthcare-seeking behavior for young people are also affected. Our findings suggest that loss of employment among household members due to the lockdown was associated with depressive symptoms among both men and women. Approximately 400 million informal sector workers in India have lost their livelihood due to COVID-19 and related lockdowns [25]; interviews with informal sector workers describe impending poverty, evictions and hunger as incomes and work opportunities are sharply curtailed [26]. Previous research has also found a link between loss of employment and SGBV, both of which likely relate to depressive symptoms during lockdown [15,16]. A recent study conducted prior to COVID-19 of mental health in India found being a woman, younger age, loss of employment, and other characteristics were associated with symptoms of depression, anxiety and stress [18].

Many women reported that they had forgone necessary medical services, which may lead to adverse secondary health outcomes and outbreaks of other diseases. Among women surveyed, most of those who did require a health service could not access them. Public transit commonly used to visit clinics was closed during lockdown, which may have affected access [2]. Challenges in accessing health services must be carefully monitored to avoid unintended secondary health crises, including outbreaks of vaccine preventable disease, stunting/undernutrition, and unintended pregnancy or poor birth outcomes [27]. While most women reported they did not require any health services, this study was conducted early in the pandemic. If lockdowns

resume or access continues to be disrupted, utilization of essential services should be monitored, and steps taken to ensure accessibility.

This study has several limitations. First there are inherent challenges in conducting surveys that are not face-to-face; mobile phone-based data collection relies on self-reported information conveyed by participants who may have challenges understanding questions, and we cannot guarantee protections for participants who may be vulnerable in their households [28]. Secondly, the representativeness of the sample may be compromised as we could only interview those with working phone numbers from the 2015–16 UDAYA survey. Our survey respondents had slightly higher educational attainment and household wealth compared to the full UDAYA cohort, suggesting that the most vulnerable from the original sample were not reachable. Third, we asked questions regarding knowledge of COVID-19 prevention behaviors, then later asked about behaviors respondents were doing. Potentially, question order nudged recall, which could explain why the proportion aware of certain behaviors was lower than those who reported implementing them. However, both the knowledge and behavior questions were based on spontaneous responses, not a list read by the interviewer, so this effect should be minimal. Lastly, our measure of mental health was very simple and self-reported, validated depression measures are necessary but challenging to collect via mobile phone interview.

Our findings suggest that early in the pandemic lockdown, there were significant knowledge gaps and secondary health effects disproportionately impacting adolescent girls and young women. To increase knowledge of symptoms and preventive behaviors, gender-sensitive behavior change campaigns should be developed, and adapted for cultural context, literacy, and accessibility. Improved access to information may lead to adoption of promoted behaviors, reducing risk of infection. Relatedly, steps to address mental health and the unintended secondary health impacts of the pandemic are required. To date, the Government of India has introduced several initiatives to address these issues, for example activating a toll-free helpline for those requiring psychosocial counseling and issuing guidelines for the sustained provision of essential health services. Government agencies are also launching special social protection initiatives. It is critical that these measures reach the most vulnerable populations, including messaging targeted to women. Longer term efforts may also be necessary to address the prolonged and potentially gendered effects of COVID-19 and ensure that health and development gains are not lost due to the pandemic, especially as India's case load has grown to one of the highest worldwide.

## Supporting information

**S1 Table. Differences in key background characteristics between respondents aged 15–19 whose number was not available, who were interviewed in the COVID-19 survey and who were not interviewed in COVID-19 survey.**
(TIF)

**S1 File. COVID-19 study questionnaire.**
(PDF)

## Acknowledgments

The authors would like to acknowledge the dedicated team at Population Council Inc. in India that collected all of these surveys and made this research happen.

## Author Contributions

**Conceptualization:** Jessie Pinchoff, KG Santhya, Rajib Acharya, Thoai D. Ngo.

**Data curation:** Shilpi Rampal.

**Formal analysis:** Jessie Pinchoff, Shilpi Rampal.

**Investigation:** Rajib Acharya, Thoai D. Ngo.

**Methodology:** KG Santhya, Corinne White, Shilpi Rampal, Rajib Acharya.

**Project administration:** KG Santhya, Corinne White, Rajib Acharya.

**Resources:** KG Santhya.

**Supervision:** Jessie Pinchoff, KG Santhya, Rajib Acharya, Thoai D. Ngo.

**Writing – original draft:** Jessie Pinchoff, Corinne White.

**Writing – review & editing:** KG Santhya, Shilpi Rampal, Rajib Acharya, Thoai D. Ngo.

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
