## [Decision Letter · Decision Letter 0]

15 Oct 2020

PONE-D-20-26559

Gender specific differences in COVID-19 knowledge, behavior and health effects among adolescents and young adults in Uttar Pradesh and Bihar, India

PLOS ONE

Dear Dr. Pinchoff,

Thank you for submitting your manuscript to PLOS ONE. After careful consideration, we feel that it has merit but does not fully meet PLOS ONE’s publication criteria as it currently stands. Therefore, we invite you to submit a revised version of the manuscript that addresses the points raised during the review process.

We look forward to receiving your revised manuscript.

Kind regards,

Kannan Navaneetham, PhD

Academic Editor

PLOS ONE

Journal Requirements:

2. Please include additional information regarding the survey or questionnaire used in the study and ensure that you have provided sufficient details that others could replicate the analyses.

For instance, the content development and validation. 

If the questionnaire it is not under a copyright more restrictive than CC-BY, please include a copy, in both the original language and English, as Supporting Information.

Reviewers' comments:

Reviewer's Responses to Questions

**Comments to the Author**

1. Is the manuscript technically sound, and do the data support the conclusions?

Reviewer #1: Yes

Reviewer #2: Yes

Reviewer #3: Yes

2. Has the statistical analysis been performed appropriately and rigorously? 

Reviewer #1: Yes

Reviewer #2: No

Reviewer #3: Yes

3. Have the authors made all data underlying the findings in their manuscript fully available?

Reviewer #1: Yes

Reviewer #2: Yes

Reviewer #3: Yes

4. Is the manuscript presented in an intelligible fashion and written in standard English?

Reviewer #1: No

Reviewer #2: Yes

Reviewer #3: Yes

5. Review Comments to the Author

Reviewer #1: This manuscript highlights the gender differences in knowledge and adoption of preventive practices against COVID-19 in two provinces in India, as well as mental health consequences that are particularly pronounced among women. The discussions contained in this paper are very important to highlight the necessity of COVID-19 response guidance with adequate gender considerations. This is especially relevant in the context of India where the cultural influence over gender norms is strong and as a country where the spread of the virus has been particularly severe. However, parts of the paper will need to be restructured in order to improve the clarity and flow of the manuscript as a narrative. In particular, it is important that challenges to health care access for women is properly conceptualized in the introduction as a study aim in order to justify its subsequent analysis. Some discrepancies and inconsistencies in language, though minor issues, are important to resolve (namely, the use of women/men vs. female/male and using a term that is broader than “depression” due to the simplistic measure).

Below are more detailed comments and recommendations:

Abstract

1. It is not immediately clear why it is highlighted that knowledge was the only predictor of practice of preventive behaviors among women, especially since we see later that it is a significant predictor for men as well. As an important key finding, the authors may consider further expanding on the implications of this key finding, e.g. that because knowledge is the only predictor for females, education programs/information campaigns are particularly important as an intervention with gender considerations.

2. Great, succinct summary of key findings in lines 39-41.

Introduction

3. Could the authors rephrase, or be more specific about what they mean by “the negative secondary effects of crises” (line 56)? Although examples of what these secondary effects include are described in subsequent paragraphs of the introduction, without an upfront definition or description, it is not immediately clear to readers what this means, or why the authors are focusing on said secondary effects as opposed to the “primary effect” of infection.

4. What do the authors mean by “structural facilitators” (line 71)?

5. The paragraph starting on line 73 needs to be restructured to have one main idea. Currently, I can identify four separate ideas: (1) There are a number of negative repercussions as a result of COVID-19 pandemic, particularly barriers to accessing health care, (2) inability to access health care is particularly problematic for women, (3) the prevalence of gender-based violence and adverse mental health outcomes may increase during lockdowns, (4) depression in response to COVID-19 has been reported among adolescents. The background information provided to readers should be focused and tailored to the authors’ stated research aims in line 87: to highlight “the gender specific variation in COVID-19 knowledge and practice of preventive behaviors, and mental health effects”.

6. If one of the study aims was to examine the adolescent and young adult population in particular, an additional paragraph justifying why is warranted. However, it could also be the case that the primary aim of the study was to examine gender differences, but using a study sample of adolescents and young adults (potentially due to reasons of feasibility / harnessing an existing cohort study), in which case this should just be acknowledged.

7. The authors may consider removing two sentences in lines 88 to 91, as it is information repeated from earlier in the introduction.

Methods

8. In line 101, the authors should specify that the original study objectives of the UDAYA study were about acquisition of assets and transition into adulthood, so it is clear that it is distinct from the objectives of the present study.

9. It is good to see that the authors explicitly state that while the survey collected information on the participants’ sex, the study would adopt the lens of gender. However, the language of gender (women/men) should then be used consistently throughout the paper.

10. The sentence “All survey responses were tabulated by gender…” in lines 140-141 should be moved to last paragraph of the methods where the statistical analysis approaches are described.

11. I do not think the reporting “feeling lonely, depressed or irritable” can be accurately labelled as experiencing depression. I suggest the authors consider using an alternate term that is broader, such as “adverse mental health effect” (as in the abstract). This comment also applies to all sections of the paper, including figures and tables.

12. It would be helpful if the word “caste” (line 150) were defined for the international reader.

13. Just as the other variables are described, the operationalization of education should also be included. In particular, an explanation for how years of education have been categorized should be provided, i.e. why is the range for the middle category of 8-9 years of education so narrow, compared to the other two categories?

14. A similar comment for religion – how were they categorized? Tables 2 and 3 state that Muslim is compared to “Hindu, other religions” whereas Table 1 suggests that the category only refers to Hindu. If other religions were combined with Hindu into one category, this needs to be specified.

15. In line 158, the authors state that logistic regression models were used, although it seems like linear probability models were used throughout the paper.

16. The authors may also consider clarifying that three models were constructed for each of the three outcomes of interest.

17. It is unclear what is meant by “Models were constructed to explore characteristics associated with reporting household employment lost due to COVID-19”, as loss of employment was not described as one of the outcome variables.

Results

18. I would suggest avoiding the language of “more likely” or “less likely” to compare the survey responses of women vs. men when they are expressed in percentages, and to reserve this for reporting probabilities. Instead, for example, the authors may simply report that fewer women (40%) were aware of the main symptoms of COVID-19, compared to men (53%).

19. The phrase “among male respondents in the male only model” (line 209) can be simplified to avoid repetition.

20. The findings about health service access (from line 220) come as a bit of a surprise to the reader, as this is not stated as a study aim, nor is this step described in the methods. If exploring challenges in accessing healthcare among women is an additional study objective, this needs to be reflected in the introduction and methods, with clear justifications for why this is a gendered issue.

Discussion

21. It may be more accurate to rephrase the finding in lines 232 and 239 (“Overall, female respondents had less accurate information of COVID-19 symptoms…”) to instead say that women were less likely to be able to identify all three COVID-19 symptoms correctly. It can then be postulated that this may be because they had less access to accurate knowledge of COVID-19, compared to men.

22. The way the sentence “This is also supported by a recent study that found differential COVID-19 risk and mortality by gender and age, suggesting most infections are among men because males are more likely to leave home for work and for socializing due to gendered social norms” (line 249-252) is structured may confuse readers, as the previous sentence was talking about knowledge of viruses, whereas the focus of this sentence jumps to risk of infection. I suggest restructuring the sentence to emphasize the similar role of gendered social norms in affecting both knowledge and infection risk.

23. It would also be beneficial to mention that this second study was also conducted in India.

24. Similar to my above comment for the abstract, authors should expand on the significance of knowledge being the only characteristic associated with adoption of behaviors among women, compared to men where sociodemographic variables also had a role to play.

25. Findings about access to health services (line 274) should be a different paragraph, as it is a separate finding from the mental health outcomes.

26. What do the authors mean by “both sets of questions were spontaneous responses only” (line 293)?

27. The limitation of the mental health measure is important and well-described.

Tables

28. (Table 1) Could the authors clarify what ‘Total’ in the top left corner refers to?

29. (Table 1) Why are the responses to only two of four preventive measures presented in particular?

30. (Table 1) Antenatal care should be written out in full since the abbreviation “ANC” has not been defined elsewhere in the manuscript.

31. (Tables 2 and 3) The reference category of ‘sex of the respondent’ should be specified.

32. (Tables 2 and 3) The abbreviation “UP” for Uttar Pradesh should be defined earlier in the manuscript.

33. (Tables 2, 3 and 4) Whether “NA” and “REF” are used should be consistent.

Reviewer #2: This area of investigation is not novel nor unique. There is a clear bias in the number of participants; more females were included in the survey. The statistical analysis section was not included in the methodology. The results were presented accurately, but the discussion section again was week and did not include comparison with findings from similar studies in the region. The study has a number of limitation which could be prevented with better planning, beside the disadvantages of using this type of phone surveys.

Reviewer #3: 1. The study shows that there is 95% responce rate in the telephonic survey. it may be further be looked into as responce rate seems to be very high (line no. 115).

2. in the line 222, Nutrition services is 51%, it may be the typo error of 52%.

3. The study may not reflect the true representation of the population covered in the selected states. As most of the respondants may mot have the phone/ Mobile no. to interview, It may be considered as the limitation of the study.

6. PLOS authors have the option to publish the peer review history of their article (what does this mean?). If published, this will include your full peer review and any attached files.

Reviewer #1: No

Reviewer #2: **Yes: **Hadil Mohammad Alahdal

Reviewer #3: **Yes: **Dilip Kumar

---

## [Author Response · Author response to Decision Letter 0]

20 Nov 2020

We have attached our specific reviewer and editor comments in an attachment titled "response to reviewers". Thank you!

---

## [Editor Report · Decision Letter 1]

3 Dec 2020

Gender specific differences in COVID-19 knowledge, behavior and health effects among adolescents and young adults in Uttar Pradesh and Bihar, India

PONE-D-20-26559R1

Dear Dr. Pinchoff,

We’re pleased to inform you that your manuscript has been judged scientifically suitable for publication and will be formally accepted for publication once it meets all outstanding technical requirements.

Kind regards,

Kannan Navaneetham, PhD

Academic Editor

PLOS ONE
---

## [Editor Report · Acceptance letter]

9 Dec 2020

PONE-D-20-26559R1 

Gender specific differences in COVID-19 knowledge, behavior and health effects among adolescents and young adults in Uttar Pradesh and Bihar, India 

Dear Dr. Pinchoff:

I'm pleased to inform you that your manuscript has been deemed suitable for publication in PLOS ONE. Congratulations! Your manuscript is now with our production department. 

Kind regards, 

on behalf of

Professor Kannan Navaneetham 

Academic Editor

PLOS ONE